Corrected: Author correction

# Forecasting cell fate during antibiotic exposure using stochastic gene expression

Nicholas A. Rossi[1,2,4], Imane El Meouche[2,3,4] & Mary J. Dunlop [1,2,3]

Antibiotic killing does not occur at a single, precise time for all cells within a population. Variability in time to death can be caused by stochastic expression of genes, resulting in differences in endogenous stress-resistance levels between individual cells in a population. Here we investigate whether single-cell differences in gene expression prior to antibiotic exposure are related to cell survival times after antibiotic exposure for a range of genes of diverse function. We quantified the time to death of single cells under antibiotic exposure in combination with expression of reporters. For some reporters, including genes involved in stress response and cellular processes like metabolism, the time to cell death had a strong relationship with the initial expression level of the genes. Our results highlight the single-cell level non-uniformity of antibiotic killing and also provide examples of key genes where cell-to-cell variation in expression is strongly linked to extended durations of antibiotic survival.

[1] Molecular Biology, Cell Biology & Biochemistry Program, Boston University, Boston, MA 02215, USA. [2] Biological Design Center, Boston University, Boston, MA 02215, USA. [3] Department of Biomedical Engineering, Boston University, Boston, MA 02215, USA. [4]These authors contributed equally: Nicholas A. Rossi, Imane El Meouche. Correspondence and requests for materials should be addressed to M.J.D. (email: mjdunlop@bu.edu)

Bacteria are killed by antibiotics, but their effect is neither instantaneous nor uniform. Rather, antibiotic exposure results in a distribution of killing times, with some bacteria succumbing to antibiotic exposure quickly while others remain viable. In population-level experiments this effect is visible in time-kill assays, which for *Escherichia coli* typically demonstrate rapid killing within a window of 1–3 h following antibiotic exposure[1]. However, survival of even a small number of cells can be critical in clinical settings, resulting in chronic infections. A well-studied example of this is bacterial persistence, where a subset of the population exists in a temporarily dormant state that renders those bacteria tolerant to antibiotics[2]. Time-kill experiments from bulk population studies result in a biphasic killing curve, with a first phase where the majority of the cells are killed rapidly, followed by a second phase where death of the remaining persister cells is much more gradual[3]. Single-cell studies have shown that these bacterial persisters can survive and regenerate populations[3,4], potentially leading to recalcitrant infections[5]. Besides the discrete persister cell state, populations of bacteria can also exhibit a continuum of resistance levels. In this case, the probability of survival under antibiotic exposure changes as a function of the expression of their stress response genes[6]. In addition to the clinical impact in chronic infections, cell-to-cell differences in antibiotic susceptibility can play a critical role in the evolution of drug resistance[7–9]. Temporal differences in survival times are important, as recent studies have shown that drug resistance can evolve rapidly under ideal, selective conditions[9,10].

Variability in gene expression arising from stochasticity in the order and timing of biochemical reactions is omnipresent, and populations of cells can leverage this noise to introduce phenotypic diversity despite their shared genetics[11]. For example, bacteria can exhibit heterogeneity in expression of stress response genes, allowing some individuals in the population to express these genes more highly, leading to survival under stress[6,8,12]. Examples of stress response machinery driven by noise include sporulation and competence pathways in *Bacillus subtilis*[13–15] and expression of lysis and lysogeny genes in λ phage[16]. In addition, we have shown that expression of the multiple antibiotic resistance activator MarA in *E. coli* is heterogeneous, which generates diverse resistance phenotypes within a population[6]. Beyond stress response, fluctuations in gene expression can inform the future outcomes of a variety of cellular states. These include examples from development, where variability in the Notch ligand Delta can effectively forecast *Drosophilia* neuroblast differentiation[17]. In addition, in cancer, human melanoma cells display transcriptional variability that determines if they resist drug treatment[18]. Additionally, knowledge of the number of lactose permease molecules in a cell can predict if individual *E. coli* induce *lac* operon genes[19]. Moreover, combining information from multiple genes may increase the capacity to forecast future cell fate, as has been shown in a yeast metabolic pathway[20].

Antibiotic-resistant infections are a major public health threat[21]. Standard population-level approaches such as those measuring minimum inhibitory concentrations mask single cell effects that can cause treatment failure[22]. Therefore, measurements revealing cell-to-cell differences in antibiotic survival times can be critical in informing how bacteria evade antibiotic treatment. Identifying genes involved in extending survival times has the potential to lead to new targets, and to reveal stepping stones in the evolution of drug resistance[9].

Here, we measure single cell killing as a function of time under antibiotic exposure. By simultaneously measuring expression of targeted genes within single cells and cell survival, we identified genes whose instantaneous expression prior to antibiotic

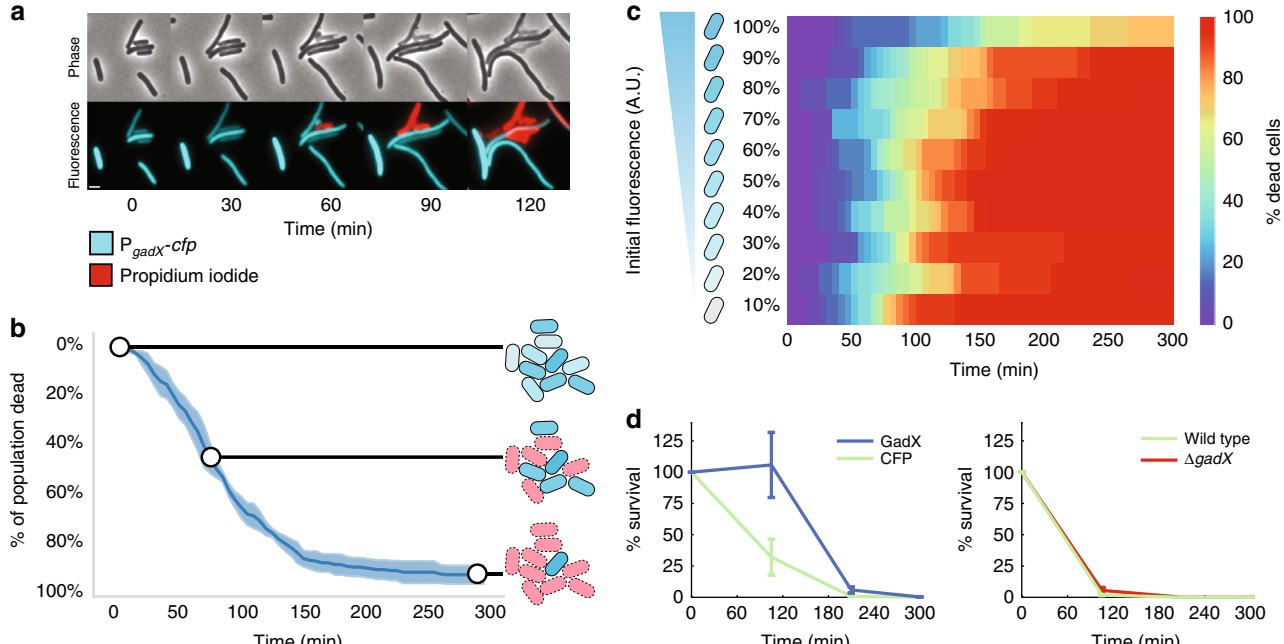

**Fig. 1** Differences in single-cell carbenicillin susceptibility. **a** Snapshots of cells demonstrate variable lethality of carbenicillin. $P_{gadX}$-*cfp* fluorescence (cyan); propidium iodide is a cell death marker (red). **b** Cellular populations die progressively after carbenicillin exposure. Line represents mean killing curve as a function of time. Shaded region represents standard deviation across five replicate microscopy positions containing ~100 cells each. Cartoon schematic demonstrates how lethality is variable among individuals within the population, but depends on initial $P_{gadX}$-*cfp* fluorescence. **c** Cells die at different times as a function of their initial $P_{gadX}$-*cfp* fluorescence. The x axis shows the cumulative percentage of dead cells at each time point. Initial fluorescence at $t = 0$ is split in deciles with equal numbers of cells in each of the ten bins along the y-axis (Supplementary Fig. 2). **d** Population-level carbenicillin killing curves for cultures containing a plasmid expressing *gadX* or *cfp*. Carbenicillin-killing curves for wild type and Δ*gadX* cultures. For both data sets, $n = 3$ biological replicates and error bars show standard error of the mean

introduction correlates with the ability to extend survival times under antibiotic exposure. To do this, we computed the mutual information between gene expression levels and the life expectancy of the cells expressing them. We found examples where gene expression can determine when the cell is likely to die, not simply if the cell is going to die. These results demonstrate the critical information contained within the stochastic expression of certain genes in their capacity to forecast cell fate. We analyze several factors, including mean expression levels, cell size, and growth rates, and reveal that both expression of certain genes and growth rate can effectively forecast cell fate, while other features are at best weakly predictive at informing survival times in the presence of antibiotics.

## Results

**_E. coli_ shows single-cell variability in carbenicillin susceptibility**. In order to quantify the relationship between stochastic gene expression and the time to _E. coli_ cell death under antibiotic exposure, we grew cells with a reporter where the promoter for a gene of interest controls expression of cyan fluorescent protein (CFP). At $t = 0$ we transferred cells with the reporter to agarose pads containing a lethal dose of carbenicillin and then monitored cell killing over time (see "Methods"). We selected carbenicillin because of its clinical relevance[23], and its bactericidal activity, which makes it straightforward to pinpoint the exact time of cell death[24]. At $t = 0$ we observed heterogeneity in gene expression, as quantified by CFP fluorescence levels (Fig. 1a). We then recorded the percentage of dead cells in the population at each time point using propidium iodide, which stains DNA if the membrane is depolarized[25].

As an example, we observed a strong relationship between gene expression levels and cell killing for the _gadX_ promoter[26]. GadX is a positively auto-regulated transcription factor that controls the expression of pH-inducible genes[27,28]. Despite the fact that all imaged cells were isogenic clones, we observed heterogeneity in $P_{gadX}$-_cfp_ expression and also in antibiotic lethality over time. The time-dependent killing curve was consistent across replicates, with cells with higher expression of $P_{gadX}$-_cfp_ at $t = 0$ surviving for longer than those with low expression (Fig. 1b).

To quantify this, we ranked the cells according to their fluorescence at $t = 0$ from low to high expression, then binned them so that each bin contained 10% of the cells. We tracked lysis of single cells over time to quantify the difference in time to death as a function of the initial fluorescence of the $P_{gadX}$-_cfp_ reporter (Fig. 1c). We found that the brightest 10% of cells, corresponding to those with the highest expression of $P_{gadX}$ prior to antibiotic exposure, survived for longer times under antibiotics than cells with lower expression (Supplementary Movie 1). To verify the role of _gadX_ expression in increasing the time to death under carbenicillin treatment we conducted additional experiments using a strain overexpressing _gadX_ and a Δ_gadX_ strain, comparing each of these with a strain with wild-type levels of _gadX_ expression (Fig. 1d). We found that cells overexpressing _gadX_ could survive carbenicillin treatment longer than those with wild-type levels; deleting _gadX_ did not alter the survival time, consistent with the heatmap data.

**Bacterial promoters vary in ability to predict carbenicillin response**. Next, we extended this analysis to include additional genes, constructing reporters for a total of 15 promoters. Our analysis included genes that covered the major branches of the gene ontology classification system for _E. coli_[29] (Supplementary Fig. 1). They include reporters for genes involved in metabolism, cell processes, cell structure, transport, information transfer, and regulation. We repeated the antibiotic exposure experiments for

each reporter and compared time of death for single cells to the initial fluorescence level of that cell. Each reporter had a unique distribution of initial fluorescence values, and ranking and dividing cells into ten equal groups gave us an unbiased way of comparing levels of gene expression given diverse means and distributions of fluorescence (Supplementary Figs. 2, 3).

We quantified the percentage of the initial population that survived for each decile (10%) of initial fluorescence over-time for all promoters (Fig. 2a). Qualitatively, we observed that certain promoters have a clear relationship between the time to cell death and the fluorescence at $t = 0$ ($P_{purA}$, $P_{inaA}$, $P_{rob}$, and $P_{gadX}$), while others die at a uniform time regardless of initial fluorescence (e.g., $P_{fis}$). These features are visible in the heatmaps showing the percentage of dead cells over time as a function of the initial fluorescence. Interestingly, not all reporters with predictive power about the time to cell death have the same characteristic shape to their heatmaps. For instance, some reporters show a negative relationship between cell death and fluorescence ($P_{inaA}$, $P_{rob}$, and $P_{gadX}$) while others show a positive relationship ($P_{purA}$). Also, in some cases there is a distinct expression level that defines a cutoff for extended survival times (top 10% of cells for $P_{gadX}$, bottom 30% of cells for $P_{purA}$). In other cases, there is a continuous relationship between fluorescence and survival ($P_{rob}$). The differences in the shape of these heatmaps may reflect the biological mechanism by which these genes offer tolerance or resistance.

For instance, expression of _purA_ is downregulated by transcription factors that increase antibiotic resistance[30]. PurA is involved in AMP synthesis, suggesting possible interplay between intracellular ATP, growth, and carbenicillin survival. _inaA_ encodes a weak acid inducible protein[31], while _gadX_ plays a regulatory role in acid resistance[27]. Interestingly, a noisy response to antibiotic stress has been shown to predict acid resistance via the _gad_ operon and the depletion of adenine nucleotides[26]. Rob is a transcription factor that is involved in a wide set of functions, from decreasing the concentration of antibiotics in cells to detoxifying oxidative stress[32]. Thus, cells with high Rob expression likely have multiple survival mechanisms.

To quantify the predictive power of each reporter in determining cell killing, we measured how the initial fluorescence decreases uncertainty about the future cell state. Because of the differences in the heatmaps, we sought to use a metric that was agnostic to the exact shape of the killing curve over time as a function of gene expression. To do this, we computed the mutual information between reporter florescence at $t = 0$ and the cellular state at each subsequent time point (see "Methods"). To provide intuition into the results, we visualized several characteristic heatmaps (Supplementary Fig. 4). If all cells are alive (as at $t = 0$) or if all cells are dead (as is the case after long periods of antibiotic exposure), the information is zero. If cells die linearly in precise proportion to their initial fluorescence, the corresponding information is a parabolic arc, where information peaks at the theoretical maximum of 1.0 bit when half the cells are dead. Variations on this pattern decrease the information. Finally, if cell death is not related to initial fluorescence, then the information is always zero.

Computing the information over time allowed us to compare the predictive power of each of the reporters (Fig. 2b). We found that the peak mutual information between initial fluorescence and cellular death varies among promoters. The peak information occurs at the point in the experiment where the initial fluorescence is the most informative about the cellular state at that time. Considering the top four promoters when ranked by peak information (Fig. 2c) ($P_{purA}$, $P_{inaA}$, $P_{rob}$, $P_{gadX}$), we found that each peak occurs at a different time point, indicating that temporal ordering of these genes may be significant in

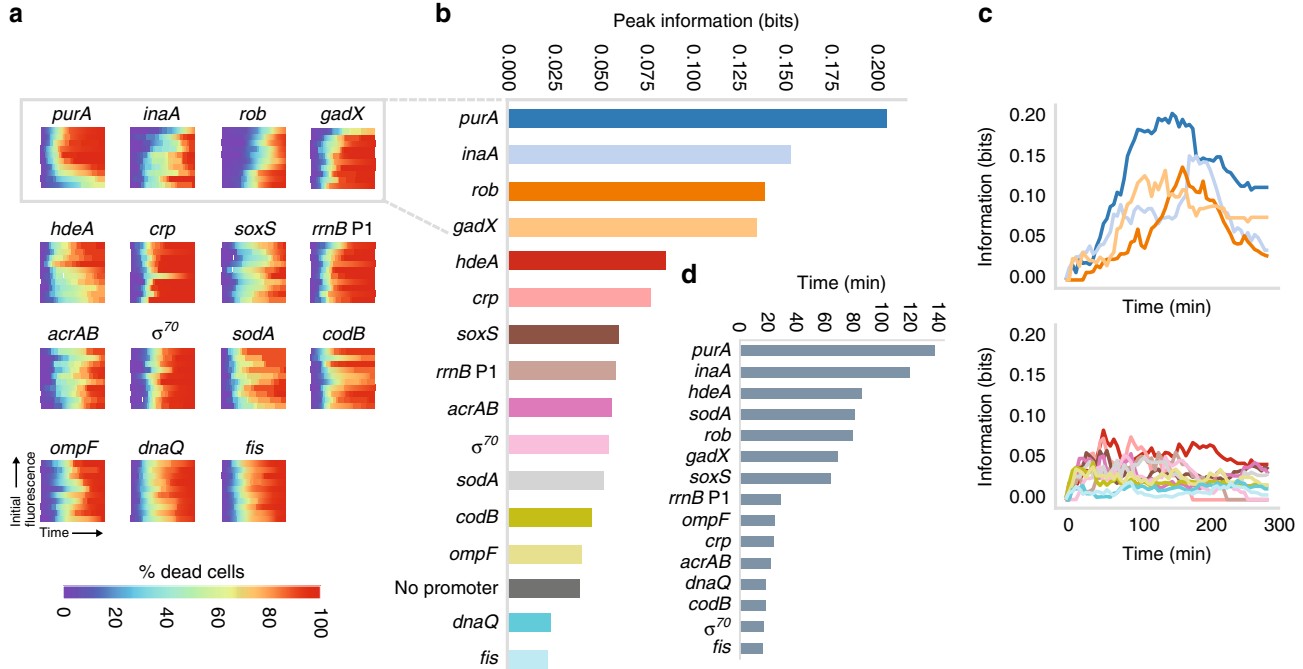

**Fig. 2** Bacterial promoters have different predictive power in the presence of carbenicillin. **a** Variable death times of cells depending on initial fluorescence. As in Fig. 1c, the x-axis shows cumulative percentage of dead cells over time and y-axis represents binned deciles according to initial fluorescence at $t = 0$. For the bins and initial fluorescence distributions for each reporter see Supplementary Fig. 2. At least five replicate microscopy positions with ~100 cells each were pooled before binning. **b** Peak mutual information between the initial fluorescence and cell fate for each reporter strain. **c** Information over time for each strain. For visual clarity, the data are divided onto two plots, one of which shows the four reporters with the highest peak information and the other showing the remaining reporter data. **d** Differences in time to reach 50% cell death between the fluorescence decile with the fastest dying cells and the decile with the slowest dying cells. Savitzky–Golay filter was used to smooth data across deciles before calculating the minimum and maximum values

determining cell killing. To provide insight into the magnitude of the peak information necessary to distinguish between random variation and clear trends in the relationship between gene expression patterns and cell killing we also conducted permutation tests on the data. In this analysis, we randomly grouped data into tenths rather than sorting by fluorescence (Supplementary Fig. 5A) and calculated the peak information (Supplementary Fig. 5B). We repeated the randomization 100 times to generate statistics for the permutation test, providing a baseline against which to judge peak information values (Supplementary Fig. S5C). The top ranked promoters all well exceeded the peak information values that would be expected due to random chance, while low peak information values typically indicate no concrete relationship between gene expression levels and killing times. As an additional control, we also included a reporter with no promoter driving *cfp* expression. As expected, the information provided by this reporter was negligible.

For each promoter, we also calculated the difference between cell killing times by measuring the difference in time to 50% cell killing between the decile where cells were killed fastest and that where they survived the longest (Fig. 2d). Cells containing $P_{purA}$ and $P_{inaA}$ reporters exhibited the greatest diversity in killing times.

We next asked if it was possible that the predictive power of a particular promoter was a result of the statistics of that promoter, not its cellular function. To do this, we calculated the correlation between the peak information and its strength (mean expression) and noise (coefficient of variation) for all reporters. We found no appreciable correlation between mean expression and peak information (Supplementary Fig. 6A), nor between the coefficient of variation and peak information (Supplementary Fig. 6B). These results show that the naive statistics of a promoter are not the reason why it is or is not predictive for cell fate.

We also computed the information between cell fate and measurements that are independent of the fluorescence, including cell size and growth rate at $t = 0$. Cell size is variable at the initial time point because cells are at different stages in the division process. We found that initial cell size has modest predictive power about survival, and the heatmap showed slightly extended survival times for smaller cells in the presence of carbenicillin (Supplementary Fig. 7). This finding on the relationship between cell size and killing time is consistent with previous research showing that the longer it has been since division, the more likely a cell is to lyse in the presence of carbenicillin[33]. However, the modesty of its predictive capacity is a testament to the relative phenotypic importance of stochastic gene expression by comparison. Although this effect is present, expression of reporters like $P_{purA}$ is far more predictive of survival than cell size. In contrast, cell growth rate is predictive of cell killing time, with slow growing cells surviving longer than fast growing ones (Supplementary Fig. 8). This agrees with recent results showing a robust correlation between growth rates and lysis rates under β-lactam antibiotic treatment[34].

**Predictive power varies by antibiotic.** Our initial experiments used carbenicillin, however, we next asked whether results on information between gene expression and cell killing were specific to this particular stressor or extended to other antibiotics. We repeated our experiments using a subset of reporters with ciprofloxacin. Ciprofloxacin is a clinically relevant antibiotic that inhibits DNA gyrase[35]. Unlike carbenicillin, it exhibits both bactericidal and bacteriostatic effects[36]. Comparing the peak mutual information between six of the promoters with the two antibiotics, we found distinct differences between their predictive power for carbenicillin and ciprofloxacin (Fig. 3a). First, $P_{gadX}$ has

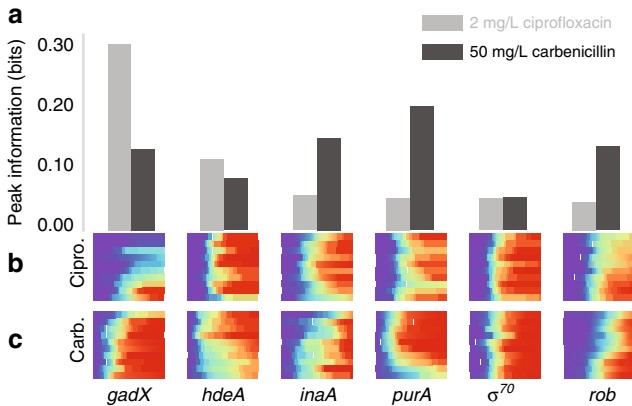

**Fig. 3** Promoters show different predictive power under ciprofloxacin versus carbenicillin. **a** Peak mutual information for ciprofloxacin and carbenicillin. **b** Variable death times for six strains under ciprofloxacin treatment. Binning on x- and y-axes performed as in Fig. 1c. **c** Variable death times under carbenicillin treatment. Data are reproduced from Fig. 2a for comparison

a peak mutual information of ~0.3 bits under ciprofloxacin treatment. This is considerably more than the information $P_{gadX}$ provides under carbenicillin. Contrary to this, $P_{purA}$, $P_{inaA}$, and $P_{rob}$ all offer more predictive power under carbenicillin stress than ciprofloxacin. $P_{\sigma70}$ and $P_{hdeA}$ offer comparable predictive power between the two antibiotics, with the constitutive promoter $P_{\sigma70}$ providing negligible capacity to forecast cell fate in either case. The heatmaps from both antibiotics also show qualitative differences in how reporters predict cell fate (Fig. 3a, b). For instance, while cells with comparatively high $P_{gadX}$ fluorescence survive well in both carbenicillin and ciprofloxacin, it is only the top decile of cells in carbenicillin that have a comparative advantage, while the upper half do in ciprofloxacin. The opposite proves true for $P_{hdeA}$ where only the lowest decile of cells have an advantage in ciprofloxacin, but there is a continuous advantage as a function of lower fluorescence in carbenicillin. Together, these results show that although gene expression may correlate well with cell death, the time to death and single-cell level killing effects can vary considerably with the type of stress.

## Discussion

We have demonstrated that differences in gene expression associated with noisy promoter activity have the potential to forecast information about the future fate of a cell. This approach allowed us to quantify how phenotypically meaningful the stochastic expression of a particular promoter is for survival in the presence of antibiotics. The promoters we selected for our reporters occupy a variety of roles (Supplementary Note). Of those we tested, reporters for stress response, metabolism, cell processes, and information transfer had the most predictive power (Supplementary Fig. 9), however, not every promoter within those classes is predictive. Interestingly, the genes that had the highest predictive power were involved in ATP synthesis and/or acid response (purA, gadX, and hdeA) in addition to regulators involved in antibiotic resistance and oxidative stress response (rob and soxS). This highlights the coupling between responses to multiple stressors such as low pH and antibiotics. It also sheds light on the need to further understand the overlapping mechanisms cells use to cope with stress.

Surprisingly, some genes known to be involved in antibiotic resistance were not detected to have a strong relationship to when the cells died. For instance, the reporter for the acrAB multidrug efflux pump had a low peak information value, despite the pump's ability to export carbenicillin[37]. This could be because promoter activity is not necessarily representative of actual proteins within the cell, where direct measurements of protein levels would provide better information[38]. Alternatively, the advantages of additional acrAB at the levels provided due to endogenous variability in promoter activity may simply be too subtle to produce a detectable phenotypic difference in these conditions. In addition, results related to cell killing times may be dependent upon imaging conditions. For example, a switch between growth in liquid cultures and the specific imaging conditions could have an impact on cell survival as a function of gene expression. To control for this, we tested $P_{gadX}$ in an additional condition where we moved cultures from LB liquid media to LB agarose pads and found results that were similar to those where the same cultures were moved to MGC pads (see "Methods"; Supplementary Fig. 10). However, there may be other conditions where these changes are important.

Interestingly, we observed some variation in the exact killing curves between the strains with the reporters (Supplementary Fig. 11A). Some strains were killed more rapidly than others, and this effect was reproducible across replicates. While the time to reach 50% dead cells does vary among strains bearing the different reporters, this time does not correlate with their peak information (Supplementary Fig. 11B). The exact source of the variation in killing curves is unclear, but it may be that the promoter copies on the reporter plasmids operate as competitive binding sites for transcription factors and other cellular machinery necessary for tolerance or resistance[39]. Further, it would be interesting to compare results for plasmid-based reporters with chromosomally-integrated constructs to identify whether copy number and competitive effects influence the timing of cell killing.

An additional question raised by this work is whether the information offered by the various reporters could be used in combination to further improve predictions about time to cell death. To tackle this problem, the notion of multivariate information could be applied to include multiple genetic reporters[40]. If multiple reporters contain nonredundant information about cell fate, it may be possible to predict the outcome of a cell based on sufficient initial data, even prior to antibiotic exposure.

By showing how life expectancy varies as a function of initial fluorescence, we demonstrate an important relationship between gene expression and time to cell death across genes of a wide range of functions. Differences in the time to death are important because they may point to underlying mechanisms by which a particular gene grants tolerance or resistance and could expose new gateways in the evolution of drug resistance. Expanding this research to include additional reporters and antibiotics has the potential to provide a global overview of how stochasticity in gene expression propagates to variability in survival times.

## Methods

**Reporter plasmids and strains.** All reporter plasmids have a kanamycin resistance cassette and a promoter transcriptionally controlling the gene for CFP. We isolated each promoter region based on annotations in the EcoCyc database[41]. The selected sequences include all known regulatory binding sites within the database. In the absence of any binding annotations, we selected a 200 bp fragment ending with the transcriptional start site. Each construct was cloned using the Gibson assembly method; the vector was either SC101 origin pBbS5k or ColE1 origin pBbE5k[6,42] (purA, inaA, hdeA, $\sigma70$, acrAB, and sodA reporters use pBbS5k; all others use pBbE5k). Primers for construct designs are listed in Supplementary Table 1. Reporter plasmids are available on AddGene (https://www.addgene.org/Mary_Dunlop/). All plasmids were transformed into E. coli strain MG1655.

***gadX* overexpression and deletion.** The gadX gene was amplified from the chromosome of E. coli MG1655 and was cloned into the medium-copy (p15A) origin vector pBbA5k[42] (Supplementary Table 1). This plasmid was transformed into E. coli MG1655.

In order to delete the *gadX* gene from the *E. coli* MG1655 chromosome, we used homologous recombination[43]. Primers with extensions homologous to the regions adjacent to the gene were used to generate the PCR products (Table S1). After recombination, the resistance marker gene was removed using the pCP20 helper plasmid encoding FLP recombinase.

**Time-lapse microscopy**. Overnight cultures were grown from single colonies in LB medium with 30 μg/ml kanamycin for plasmid maintenance. From these cultures, a 1:100 dilution was used to inoculate fresh LB with kanamycin. Cultures were incubated for 4 h at 37 °C with shaking. Cells were then diluted 3:10 in M9 minimal medium containing 0.2% glycerol, 0.01% casamino acids, 0.15 μg/ml biotin, and 1.5 μM thiamine (which we denote MGC medium). Cells were then placed on 1.5% MGC low melting temperature agarose pads containing either 50 μg/ml carbenicillin or 2 μg/ml ciprofloxacin along with 10 μg/ml propidium iodide. For the LB agarose pad experiments (Supplementary Fig. 10), cells were diluted 3:10 in LB and then placed on 1.5% LB low melting agarose pads containing 50 μg/ml carbenicillin along with 10 μg/ml propidium iodide. For all conditions, cells were imaged at 100× using a Nikon Instruments Ti-E microscope. The elapsed time between adding cells to the pads and the initial imaging time point ($t = 0$) was no more than 15 min. The temperature of the microscope chamber was held at 32 °C for the duration of the movies. Images were taken after every 5 min for 5 h for at least five pad positions per strain, with each image containing ~100 cells.

For the heatmaps and information calculations we used microscopy settings that optimized our ability to visualize variation in reporter expression (Supplementary Fig. 2). As a result, we used different imaging exposure times for each of the reporters. When calculating mean and coefficient of variation of reporters, we used data from a separate experiment with identical imaging conditions to allow for comparison across all strains (Supplementary Fig. 3).

**Image analysis**. We tracked cell death using a combination of custom MATLAB scripts and manually scanning through the movies to locate the time of death. Our MATLAB scripts adapted the SuperSegger software for the initial segmentation[44]. We used propidium iodide fluorescence as well as other visual markers (loss of contrast in phase images, compromises to the cell wall) to ascertain the moment of cell death (Supplementary Fig. 12). It is important to note that ciprofloxacin inhibits DNA gyrase, which might lead to cell death without or before membrane depolarization. For these reasons, in addition to propidium iodide, we also relied on other visual markers for identifying the timing of cell death (Supplementary Fig. 12). We note that ciprofloxacin induces TisB, a toxin involved in membrane depolarization and persister induction[45], therefore, there may be cases where propidium iodide staining could mischaracterize cell death.

Growth rates were estimated using increases in cell size during the first five frames of the movie (20 min), as identified with SuperSegger[44]. We selected this interval because it is rare for cells to die during this initial period (Fig. 2a).

**Population-level survival rates**. Overnight cultures were diluted 1:100 in LB medium with 30 μg/ml kanamycin for plasmid maintenance. At 3 h, when cultures reached exponential phase, aliquots were diluted and plated on LB agar in order to determine the number of colony forming units before antibiotic exposure. We then added 50 μg/ml carbenicillin and cultures were incubated for 5 h. After 5 h, cells were diluted and plated on LB agar in order to determine the number of colony forming units following antibiotic exposure.

**Computing mutual information**. We computed the mutual information between cellular state (alive or dead) at time $t$ ($x_t$) and the initial fluorescence of that cell for a given promoter ($y$).

$$I(x_t, y) = H(x_t) - H(x_t | y)$$

We computed the entropy of $x_t$ from the binary entropy formula. $p(x_t)$ is computed as a fraction of cells dead at time $t$, across all initial fluorescence values for that time point.

$$H(x_t) = -p(x_t) \log_2 p(x_t) - (1 - p(x_t)) \log_2 (1 - p(x_t))$$

Finally, the conditional entropy is computed for a given initial fluorescence level ($y_i$). Where $i$ is one of the ten deciles of initial fluorescence (Supplementary Fig. 2). We then average the conditional entropy across all fluorescence bins to calculate the average conditional entropy over time. We optimized the number of bins given the number of individual cells analyzed for each strain (~500 cells)[46], however the general trends are not sensitive to the exact bin number (Supplementary Fig. 13). Analysis was conducted using custom Python scripts.

$$H(x_t | y) = \frac{1}{n} \sum_{i=1}^{n} -p(x_t | y_i) \log_2 p(x_t | y_i) - (1 - p(x_t | y_i)) \log_2 (1 - p(x_t | y_i))$$

**Permutation test**. To assess the significance of the peak information values, we compared results based on information calculations conducted with data sorted based on fluorescence at $t = 0$ with those where the data order was randomized. For the randomized data, we divided each data set into tenths and then conducted the information calculations. We repeated this process 100 times to calculate statistics across many instances of the randomization.

**Gene ontology**. To map our data to the functions of each gene, we looked up the role of each gene from the multifunctional classification scheme[29,41]. We pooled the categories of information transfer and regulation, as they were entirely overlapping for the promoters we selected.

**Reporting summary**. Further information on research design is available in the Nature Research Reporting Summary linked to this article.

## Data availability
Raw data sets are available at: https://gitlab.com/dunloplab/forecasting-cell-fate. Any other data are available from the authors upon reasonable request.

## Code availability
Custom code for data analysis is available at: https://gitlab.com/dunloplab/forecasting-cell-fate.

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

## Acknowledgements

We thank Pankaj Mehta for helpful discussions. This work was supported by the National Institutes of Health grant R01AI102922 and National Science Foundation grant 1347635.

## Author contributions

N.A.R. performed the experiments, wrote the data analysis software, and analyzed the results; I.E.M. performed the experiments and conducted the data analysis; M.J.D. contributed to the analysis software. All the authors wrote the manuscript.

## Additional information

**Competing interests:** The authors declare no competing interests.

