## [Peer Review File · Communications Biology]

Reviewers' comments:

Reviewer #1 (Remarks to the Author):

In this manuscript, Rossi et al. analyse natural variability in bacterial gene expression and its correlation with the time of cell during antibiotic exposure. The team constructed 15 different promoter-CFP vectors and tracked the initial fluorescence of individual bacteria of each strain and correlated this with time of cell death by microscopy. They show that initial activity of some, but not all, promoters correlates to the time it takes for a cell to be killed.

1. The manuscript analyses 15 promoter-CFP fusions. Do the authors know how leaky the used promoter is and what their results would be when analysing a strain containing the CFP-encoding vector without promoter present?

2. The manuscript highlights an interesting phenomenon; that heterogeneity is linked with antibiotic tolerance. In its current version it mainly highlights a correlation and does not attempt to show a molecular mechanism behind this correlation, e.g. *gadX* expression and carbenicillin tolerance. The relevance of the used technique to understand biology would be greatly enhanced if the authors would attempt to validate the link between *gadX* and carbenicillin. For example, do *gadX* over-expressing bacteria survive carbenicillin treatment better over the entire population? Or are bacteria with initial low vs. high *gadX* expression (after cell sorting) killed differently by carbenicillin?

3. the methods have been written down quite compact. Reproducibility of the experiments would be enhanced if the authors would include:

A. the name of the vector backbone

B. the method of integrating the promoters into the vector backbone

C. deposition of the vectors (e.g. in addgene)

D. more extensive explanation of the steps taken in the MATLAB scripts

E. the method for propidium iodide addition during the experiment (timing, concentration?)

4. The authors have written in the introduction: 'leading to recalcitrant infections'. Although there is much circumstantial evidence, I am not sure it has been proven that *E. coli* persisters lead to recalcitrant infections.

5. The authors introduce the topic of persisters and antibiotic tolerance. Later in the manuscript they switch to 'resistance'. Please continue to use the term 'tolerance', since the both are biologically different. (e.g. in Results: 'The differences in the shape of these heat maps may reflect the biological mechanism by which these genes offer resistance' and in discussion: 'and other cellular machinery necessary for resistance')

6. Please fix reference 30

Reviewer #2 (Remarks to the Author):

This manuscript describes how differences in expression of 15 genes (as measured by *cfp* reporters) relate to cell death (as measured by propidium iodide fluorescence) over the course of treatment with carbenicillin or ciprofloxacin. The authors find that survival times in carbenicillin are conditional on initial expression levels of four of the 15 genes (*purA*, *inaA*, *rob*, and *gadX*). The authors rule out several competing explanations that could explain cell survival independent of the initial expression, such as the cell size, the mean promoter expression, and the coefficient of variation of the promoter expression. The authors establish that initial expression levels also help forecast survival under treatment from ciprofloxacin, suggesting that the ability for the variability of gene expression to inform survival time estimates is the norm. The authors conclude by commenting on the comparative lack of importance for the expression of known antibiotic resistance genes like *acrAB*, and the possibility of building models to forecast cell survival based on the single-cell expression of multiple genes.

I find the manuscript is well-written and presented clearly. The figures are visually appealing and appropriately informative. I also find the question of whether cell survival can be predicted from initial single-cell expression relevant both as a clinical matter (for combating antibiotic tolerance and resistance) and as a basic science matter (for better understanding how organisms leverage stochasticity).

However, I feel the manuscript lacks essential context to (i) make the novelty of the work clear, (ii) interpret the results either in terms of a model or biological mechanism, and (iii) discuss possible confounding factors along with their plausibility. Thus, I cannot recommend this manuscript for publication without the specific comments being addressed.

(a) It is not clear to me exactly why the forecasting of cell survival is being studied. The first two paragraphs of the paper discuss the problem of persistence and the manifestations of cell variability, but it is left to the reader to infer what new knowledge is gained by studying cell survival in single cell experiments. I presume that the survival time has not been studied in single cells before, and that this method offers advantages over others that have been published previously. As a reader, I would like to understand what those advantages are.

(b) The paper needs more interpretation of the biological significance of the results. Based on the role of the various genes tested and the mechanism of action of the antibiotics, what can be said about the biological mechanisms driving cell death? Is the predictive power of particular genes in favor of one mechanism or another? Are cells that die early capitulating by the same mechanism as those that die later, or are there multiple mechanisms at work? When I read "The differences in the shape of these heatmaps may reflect the biological mechanism by which these genes offer resistance," I was hoping for more interpretation of this statement.

(c) I'm a bit unclear as to exactly how the cfp reporter corresponds to the expression of the gene. My understanding is that the cfp is on a plasmid under the control of the same promoter as the gene in the E. coli genome. Thus whenever the plasmid copy gets expressed, cfp is also transcribed, which increases fluorescence. Because plasmid expression may be suppressed in favor of chromosomal expression, I wonder if there would be fewer confounding factors if the cfp construct were integrated into the genome. As the authors point out in the discussion Fig. S7A shows a 'lag' time for certain strains before they start dying, and I wonder if it points to a confounding factor.

(d) I propose using a permutation test of the data, i.e., randomly grouping the time series for each reporter into tenths and calculating the distribution of peak mutual information. If the observed peak value is significantly higher than the randomized peak value, then there is evidence of some non-random structure to the data. This could help provide a statistical basis for identifying genes whose initial expression informs cell survival.

(e) When testing the coefficient of variation, the $r=0.36$ is termed "not significant." Usually when I encounter "significant" in this context, I think it means "statistically significant." Is this the case here? If so, what are the error bars (or p-value) on the 0.36? Furthermore, The histogram plotted in Fig. S2 for purA is inconsistent with it having a mean less than inaA in Fig. S4. Also in Fig. S2, there seems to be a missing 100% indicator for crp. My suspicion is that a minimal amount of variability is necessary, but not sufficient, to be able to provide information regarding cell survival.

(f) How variable are the measurements of fluorescence? It may make sense to ensure that bins of fluorescence are somewhat wider than the measurement of intensity. If the genes are really predictive of cell survival, then the binning shouldn't make a huge difference in the Results.

(g) Consider using Roman numerals of lower-case letters instead of capital letters in Fig. S3 since capital letters are reserved for figure panels.

(h) Check Refs. 1, 3, 9, 18, 20, 21, 26, 30 and 34 to ensure that all necessary information is Present.

(i) While reading this paper I was reminded of a relatively recent paper that studied single-cell variability in the context of melanoma resistance to cancer treatments:
<https://www.nature.com/articles/nature22794>.

Reviewer #3 (Remarks to the Author):

In this work Rossi and coworkers have used fluorescent transcriptional reporters combined with time lapse microscopy to study how cell to cell variation in gene expression is related to survival duration for an isogenic population of *E. coli* in the presence of lethal doses of antibiotics. The authors tested a set of 15 promoters and showed that for some promoters the expression level is predictive of survival duration. While limited in terms of the small selection of promoters tested, this is an interesting paper and a useful as a proof of principle to show that the expression level of some promoters correlates with the duration which cells survive antibiotic exposure.

My primary concern, which is not made very clear in the text, is that cells are not in a steady state at $t=0$, since according to the methods they have just been diluted from nutrient rich (LB) to poor media (M9) then moved from liquid culture to M9 agarose pads. This is likely to cause major changes to gene expression. Since β -lactam killing is very dependent on growth rate, cells which take longer to adapt to the new environment would likely have longer lag times and increased survival. Indeed, the most predictive promoters are related to metabolism and stress sensing. While the expression of some of the promoters observed in this study may be predictive under these specific experimental conditions, they may not be predictive under killing when antibiotics are added to exponentially growing cells without making any other environmental changes. This caveat should be made clear in the text. The authors should also detail in the methods section how much time elapsed between adding the cells to the agarose pad and measuring the $t=0$ fluorescence.

To determine how much of the correlation between promoter activity and survival time could be explained by differences in lag times and growth rates, it would be useful to also analyze this data, which can be extracted from the time-lapse imaging using the supersegger software already used in this report. How predictive was growth rate and lag time compared to the fluorescence reporters?

My other main concern is the assay for cell death: propidium iodide which stains DNA in cells with depolarized membranes. For carbenicillin treatment which kills by lysis this seems like a reasonable readout of cell death, however I am not convinced that it provides a reasonable readout in the case of ciprofloxacin treatment which kills cells by chromosome fragmentation. These cells may have experienced irrecoverable DNA damage long before a loss of membrane PMF. Fluoroquinolone treatment has also been shown to induce a persister state in some cells via expression of the toxin TisB which depolarizes the membrane [Dörr, T., Vulić, M., & Lewis, K. (2010). Ciprofloxacin causes persister formation by inducing the TisB toxin in *Escherichia coli*. *PLoS Biology*, 8(2), e1000317].

Despite surviving the treatment, these cells would presumably appear as 'dead' by propidium iodide stain. The limitations of this assay should be made clear in the text.

Other points:

The fluorescence units in Fig S2 and S4 are not the same – it would be preferable to use the same scale to be able to compare the data in the two figures. The fluorescence units in Fig. S2 should also be the same for all plots – it is currently misleading: *purA* appears to have the highest fluorescence signal in Fig. S2, but among the lowest in Fig. S4.

More information about why the specific 15 promoters were chosen would be useful, and more context about their function would be useful.

It would be useful to include cell size on the Fig 2 B to make the comparison of how predictive it is compared to the various promoter expression reporters.

In the introduction the authors say "We analyze a variety of factors, including cell size and mean expression levels, and reveal that expression of certain genes can effectively forecast cell fate, while many other features are at best weakly predictive at informing survival times in the presence of antibiotics." However, in this report they do not look at any other factors apart from cell apart from

cell size and promoter expression, but 'many other features' implies otherwise. This sentence should be re-phrased.

Point-by-Point Response to Reviewers

N. A. Rossi, I. El Meouche, and M. J. Dunlop
Manuscript number: COMMSBIO-19-0109-T

We appreciate the reviewers' comments and have addressed all their concerns in the revised manuscript. A complete list of changes and detailed responses to the comments follows.

Reviewer #1 (Remarks to the Author):

In this manuscript, Rossi et al. analyse natural variability in bacterial gene expression and its correlation with the time of cell during antibiotic exposure. The team constructed 15 different promoter-CFP vectors and tracked the initial fluorescence of individual bacteria of each strain and correlated this with time of cell death by microscopy. They show that initial activity of some, but not all, promoters correlates to the time it takes for a cell to be killed.

1. The manuscript analyses 15 promoter-CFP fusions. Do the authors know how leaky the used promoter is and what their results would be when analysing a strain containing the CFP-encoding vector without promoter present?

We thank the reviewer for this suggestion. We developed a CFP-encoding vector without the promoter and used it to address this question. We found that expression was very low and that with this reporter there was no relationship between fluorescence and cell killing times. We include the no promoter control in comparisons of peak information (comparison to existing data), the permutation tests (new), and bin size effect tests (new). The data are also included in the histograms and scatter plots for comparison to other promoters.

→ Results, Fig. 2B, Figs. S2, S3, S5, S6, S11, S13

2. The manuscript highlights an interesting phenomenon; that heterogeneity is linked with antibiotic tolerance. In its current version it mainly highlights a correlation and does not attempt to show a molecular mechanism behind this correlation, e.g. gadX expression and carbenicillin tolerance.

The relevance of the used technique to understand biology would be greatly enhanced if the authors would attempt to validate the link between gadX and carbenicillin. For example, do gadX over-expressing bacteria survive carbenicillin treatment better over the entire population? Or are bacteria with initial low vs. high gadX expression (after cell sorting) killed differently by carbenicillin?

*The reviewer raises an interesting question. We conducted additional validation experiments to test the link between GadX levels and carbenicillin survival. To do this, we performed two additional experiments. In the first, we tested the effect of gadX overexpression by transforming wild type *E. coli* with a plasmid containing gadX and compared it to a strain with the same plasmid containing only the gene for cyan fluorescent protein. We then measured survival over time in the presence of a lethal dose of carbenicillin. In a second experiment, we compared wild type *E. coli* to a Δ gadX*

strain. We found that the strain overexpressing gadX survived longer than cells expressing cfp alone. There were no major differences in survival time between the wild type and Δ gadX strains. These results are consistent with the data from the gadX heat map (Fig. 1C), where cells with high expression of gadX have increased survival times in the presence of carbenicillin. These results provide strong support for the link between high levels of gadX expression and carbenicillin survival.

→ See Fig. 1D, Results, Methods

3. the methods have been written down quite compact. Reproducibility of the experiments would be enhanced if the authors would include:

- A. the name of the vector backbone
- B. the method of integrating the promoters into the vector backbone
- C. deposition of the vectors (e.g. in addgene)
- D. more extensive explanation of the steps taken in the MATLAB scripts
- E. the method for propidium iodide addition during the experiment (timing, concentration?)

A, B. We added the name of the backbone and the method of generating the constructs.

C. We are happy to deposit the plasmids on AddGene prior to publication. (We do this generally for the lab's papers: https://www.addgene.org/Mary_Dunlop/). We will work with the journal to identify their preferred timing for plasmid submission.

D. We have expanded the description. However, we are also happy to share the code upon publication and typically do this on our GitLab page (<https://gitlab.com/dunloplab>). We have included an availability statement and will work with the journal to include this (once the revision is complete so that the code is finalized).

E. The concentration of propidium iodide used was included in the Methods section. As the propidium iodide was incorporated in the pads when they were made; we have clarified this in the Methods section to avoid confusion.

→ See Methods

4. The authors have written in the introduction: 'leading to recalcitrant infections'. Although there is much circumstantial evidence, I am not sure it has been proven that E. coli persisters lead to recalcitrant infections.

We thank the reviewer for this comment and have clarified the text accordingly.

→ See Introduction

5. The authors introduce the topic of persisters and antibiotic tolerance. Later in the manuscript they switch to 'resistance'. Please continue to use the term 'tolerance', since the both are biologically different. (e.g. in Results: 'The differences in the shape of these heat maps may reflect the biological mechanism by which these genes offer resistance' and in discussion: 'and other cellular machinery necessary for resistance')

This is an excellent point, as there has been confusion about these terms in the field. We clarified this in our text based on the recent consensus statement (Balaban et al. Nature Reviews Microbiology, 2019) regarding tolerance, persistence, and resistance. The mechanisms we study include both tolerance and resistance so in the text we use both words in places where we are discussing general findings, and refer to more specific resistance mechanisms when we are discussing individual genes.

→ See Introduction, Results, Discussion

6. Please fix reference 30

Fixed

Reviewer #2 (Remarks to the Author):

This manuscript describes how differences in expression of 15 genes (as measured by cfp reporters) relate to cell death (as measured by propidium iodide fluorescence) over the course of treatment with carbenicillin or ciprofloxacin. The authors find that survival times in carbenicillin are conditional on initial expression levels of four of the 15 genes (purA, inaA, rob, and gadX). The authors rule out several competing explanations that could explain cell survival independent of the initial expression, such as the cell size, the mean promoter expression, and the coefficient of variation of the promoter expression. The authors establish that initial expression levels also help forecast survival under treatment from ciprofloxacin, suggesting that the ability for the variability of gene expression to inform survival time estimates is the norm. The authors conclude by commenting on the comparative lack of importance for the expression of known antibiotic resistance genes like acrAB, and the possibility of building models to forecast cell survival based on the single-cell expression of multiple genes.

I find the manuscript is well-written and presented clearly. The figures are visually appealing and appropriately informative. I also find the question of whether cell survival can be predicted from initial single-cell expression relevant both as a clinical matter (for combating antibiotic tolerance and resistance) and as a basic science matter (for better understanding how organisms leverage stochasticity).

However, I feel the manuscript lacks essential context to (i) make the novelty of the work clear, (ii) interpret the results either in terms of a model or biological mechanism, and (iii) discuss possible confounding factors along with their plausibility. Thus, I cannot recommend this manuscript for publication without the specific comments being addressed.

(a) It is not clear to me exactly why the forecasting of cell survival is being studied. The first two paragraphs of the paper discuss the problem of persistence and the manifestations of cell variability, but it is left to the reader to infer what new knowledge is gained by studying cell survival in single cell experiments. I presume that the survival time has not been studied in single cells before, and that this method offers advantages over others that have been published previously. As a reader, I would like to understand what those advantages are.

We thank the reviewer for this comment. We have updated the text in the Introduction to clarify the importance of the work.

→ See Introduction

(b) The paper needs more interpretation of the biological significance of the results. Based on the role of the various genes tested and the mechanism of action of the antibiotics, what can be said about the biological mechanisms driving cell death? Is the predictive power of particular genes in favor of one mechanism or another? Are cells that die early capitulating by the same mechanism as those that die later, or are there multiple mechanisms at work? When I read “The differences in the shape of these heatmaps may reflect the biological mechanism by which these genes offer resistance,” I was hoping for more interpretation of this statement.

We appreciate this point and we have added more discussion about the biological significance in the Results and Discussion in the text. In addition, we include a new section in Supplementary Text that goes into greater detail on the role of the genes we studied.

→ See Results, Discussion, Supplementary Text

(c) I’m a bit unclear as to exactly how the cfp reporter corresponds to the expression of the gene. My understanding is that the cfp is on a plasmid under the control of the same promoter as the gene in the E. coli genome. Thus whenever the plasmid copy gets expressed, cfp is also transcribed, which increases fluorescence. Because plasmid expression may be suppressed in favor of chromosomal expression, I wonder if there would be fewer confounding factors if the cfp construct were integrated into the genome. As the authors point out in the discussion Fig. S7A shows a ‘lag’ time for certain strains before they start dying, and I wonder if it points to a confounding factor.

The reviewer is correct that cfp is on a plasmid under the control of the same promoter that is on the genome. As described in point (e) below, it is challenging to visualize the weaker reporters even on plasmids so we were hesitant to conduct experiments on the chromosome. However, we agree that this is a relevant point so we have noted this in the Discussion as part of a description of potential confounding effects.

→ See Discussion

(d) I propose using a permutation test of the data, i.e., randomly grouping the time series for each reporter into tenths and calculating the distribution of peak mutual information. If the observed peak value is significantly higher than the randomized peak value, then there is evidence of some non-random structure to the data. This could help provide a statistical basis for identifying genes whose initial expression informs cell survival.

We thank the reviewer for this helpful suggestion. We have conducted the permutation test as the reviewer describes, randomly grouping the data into tenths and calculating the

peak mutual information. By generating many randomized sets, we were able to calculate statistics on the level of information present in each data set. We compare these to the peak mutual information associated with the data sorted by fluorescence.

→ See Fig. S5, Results, Methods

(e) When testing the coefficient of variation, the $r=0.36$ is termed “not significant.” Usually when I encounter “significant” in this context, I think it means “statistically significant.” Is this the case here? If so, what are the error bars (or p-value) on the 0.36? Furthermore, The histogram plotted in Fig. S2 for *purA* is inconsistent with it having a mean less than *inaA* in Fig. S4. Also in Fig. S2, there seems to be a missing 100% indicator for *crp*. My suspicion is that a minimal amount of variability is necessary, but not sufficient, to be able to provide information regarding cell survival.

Regarding the “not significant” discussion: We have revised the text to avoid implying statistical significance. We also note that in the course of the revisions, we repeated all experiments measuring the mean and coefficient of variation of reporters and included new data sets, such as the no promoter control, and have recalculated the Pearson correlation values so the numbers are now $r = 0.06, 0.13, 0.13$. The larger point about statistical significance is also partially addressed by the new permutation test results (Fig. S5), discussed in point (d) above.

→ See Results, Figs. S6, S11

Regarding the histograms: This point was raised by both Reviewers 2 and 3 and we realize that our treatment of these data was quite unclear in the original version of the manuscript. We have updated the revised version accordingly. We used two different conditions for microscopy imaging in this study. In one, we set all exposure times to be identical across all reporters measured. This allows for direct comparison of values like the mean and coefficient of variation. However, the downside of this approach is that the reporters span a broad range of expression levels so there is no single exposure time that provides clear visualization of all reporters. Therefore, in experiments with constant exposure times a subset of the reporters appear at essentially background levels. Thus, in the second set of experiments we adjusted microscopy exposure times so that all reporters were above background levels so that we could visualize cell-to-cell heterogeneity. These experiments are used to generate the heatmaps of time to cell death. However, it is not fair to compare expression of one promoter to the next since microscopy imaging conditions are different. We now include histograms for both the constant exposure times (Fig. S3) and the optimized exposure times (Fig. S2). We also explain this distinction in the Methods and the relevant figure captions list which data set was used.

→ See Figs. S2, S3

*Regarding missing 100% indicator for *crp*: Fixed.*

→ See Fig. S2

(f) How variable are the measurements of fluorescence? It may make sense to ensure that bins of fluorescence are somewhat wider than the measurement of intensity. If the genes are really predictive of cell survival, then the binning shouldn't make a huge difference in the Results.

To address this, we used different numbers of bins: $N_{bins} = 10$, which is the number used in the main figures in the manuscript and also $N_{bins} = 5$ and $N_{bins} = 15$. Although there are some variations in the exact value for peak information with different numbers of bins, the trends are largely consistent across these three different analysis conditions. Coupled with the permutation test data (Fig. S5), these data sets suggest that reporters associated with high information are predictive of survival.

→ See Fig. S13, Methods (“Computing mutual information” section)

(g) Consider using Roman numerals of lower-case letters instead of capital letters in Fig. S3 since capital letters are reserved for figure panels.

Changed. Thank you for the suggestion.

→ See Fig. S4 in updated numbering in the revised manuscript

(h) Check Refs. 1, 3, 9, 18, 20, 21, 26, 30 and 34 to ensure that all necessary information is Present.

Fixed

(i) While reading this paper I was reminded of a relatively recent paper that studied single-cell variability in the context of melanoma resistance to cancer treatments:
<https://www.nature.com/articles/nature22794>.

We are flattered that our work reminded the reviewer of this interesting paper. We added this among the examples that we give on phenotypic variability in the Introduction.

Reviewer #3 (Remarks to the Author):

In this work Rossi and coworkers have used fluorescent transcriptional reporters combined with time lapse microscopy to study how cell to cell variation in gene expression is related to survival duration for an isogenic population of E. coli in the presence of lethal doses of antibiotics. The authors tested a set of 15 promoters and showed that for some promoters the expression level is predictive of survival duration. While limited in terms of the small selection of promoters tested, this is an interesting paper and a useful as a proof of principle to show that the expression level of some promoters correlates with the duration which cells survive antibiotic exposure.

My primary concern, which is not made very clear in the text, is that cells are not in a steady state at $t=0$, since according to the methods they have just been diluted from nutrient rich (LB) to poor media (M9) then moved from liquid culture to M9 agarose pads. This is likely to cause major changes to gene expression. Since β -lactam killing is very dependent on growth rate, cells which take longer to adapt to the new environment would likely have longer lag times and increased survival. Indeed, the most predictive promoters are related to metabolism and stress sensing. While the expression of some of the promoters observed in this study may be predictive under these specific experimental conditions, they may not be predictive under killing when antibiotics are added to exponentially growing cells without making any other environmental changes. This caveat should be made clear in the text. The authors should also detail in the methods section how much time elapsed between adding the cells to the agarose pad and measuring the $t=0$ fluorescence.

*This is an excellent point and we agree that it should be clear that the imaging workflow represents a change in the conditions which could have consequences for antibiotic killing. We have addressed this comment in two ways. First, we conducted an additional experiment where we transferred cells grown in LB liquid medium to LB agarose pads (rather than MGC pads) and then repeated our analysis of cell killing times. We conducted this analysis for the *gadX* promoter and compared results to the MGC pad case. We observed similar outcomes for both experiments, with the top decile of cells surviving longer than those with lower fluorescence. Second, we have also updated the text to make it clear that there is the potential for non-steady state behavior and growth dependent effects that could confound results. We have also updated the Methods to describe how much time elapses between when cells are added to the pad and $t = 0$.*

→ See Fig. S10, Discussion, Methods (“Time-lapse microscopy” section)

To determine how much of the correlation between promoter activity and survival time could be explained by differences in lag times and growth rates, it would be useful to also analyze this data, which can be extracted from the time-lapse imaging using the supersegger software already used in this report. How predictive was growth rate and lag time compared to the fluorescence reporters?

*This is an interesting question. We addressed this by re-analyzing the data for the killing experiments for the *gadX* promoter, this time calculating growth rates during the first 20 mins. We observed a correlation between growth and killing where cells that grow more slowly live longer, which is consistent with results in the literature (e.g. Lee et al. 2018 PNAS, DOI:10.1073/pnas.1719504115). It is important to note that in these experiments growth rate measurements are potentially influenced by the antibiotics, as the cells face antibiotics directly when they are loaded onto the pads. The growth is then altered, as cells go on to filament or lyse.*

*Regarding the point about lag time: We agree with this point and to avoid changing media and extending lag times as described above we repeated the killing experiment for cells containing the *gadX* reporter on pads containing LB. This way the cells should*

experience less of an adaptation period due to a change in media, though we do note the potential influence of these non-steady state effects, as described above.

→ See Fig. S8, Results, Fig. S10

My other main concern is the assay for cell death: propidium iodide which stains DNA in cells with depolarized membranes. For carbenicillin treatment which kills by lysis this seems like a reasonable readout of cell death, however I am not convinced that it provides a reasonable readout in the case of ciprofloxacin treatment which kills cells by chromosome fragmentation. These cells may have experienced irrecoverable DNA damage long before a loss of membrane PMF. Fluoroquinolone treatment has also been shown to induce a persister state in some cells via expression of the toxin TisB which depolarizes the membrane [Dörr, T., Vulić, M., & Lewis, K. (2010). Ciprofloxacin causes persister formation by inducing the TisB toxin in Escherichia coli. PLoS Biology, 8(2), e1000317].

Despite surviving the treatment, these cells would presumably appear as ‘dead’ by propidium iodide stain. The limitations of this assay should be made clear in the text.

This is a great point. We agree with the reviewer on the limitations of the propidium iodide staining and have added discussion in the Methods. In addition to the propidium iodide staining, we also relied on other physical properties such as loss of contrast and changes in cell morphology when classifying cell outcomes. We now include representative images to make it clear what kind of visual properties we associated with cell death (Fig. S12).

→ See Fig. S12, Methods (“Image analysis” section)

Other points:

The fluorescence units in Fig S2 and S4 are not the same – it would be preferable to use the same scale to be able to compare the data in the two figures. The fluorescence units in Fig. S2 should also be the same for all plots – it is currently misleading: purA appears to have the highest fluorescence signal in Fig. S2, but among the lowest in Fig. S4.

Please see response to Reviewer 2, point (e) regarding histograms. Both Reviewers 2 and 3 raised this point and we realized that our original presentation was quite confusing. To resolve this, we have added additional experimental data and clarified the Methods, Results, and Figure captions.

→ See Figs. S2, S3

More information about why the specific 15 promoters were chosen would be useful, and more context about their function would be useful.

To address this comment we have now detailed the biological functions of the genes studied and their significance in the Results, Discussion, and in new Supplementary Text.

→ *See Results, Discussion, Supplementary Text*

It would be useful to include cell size on the Fig 2 B to make the comparison of how predictive it is compared to the various promoter expression reporters.

We prefer to keep the different data types (fluorescence, cell size, and cell growth rate) separate to avoid confusion so we have not combined these data sets. However, we now include text in the caption regarding this comparison.

→ *See Fig. S7 caption*

In the introduction the authors say “We analyze a variety of factors, including cell size and mean expression levels, and reveal that expression of certain genes can effectively forecast cell fate, while many other features are at best weakly predictive at informing survival times in the presence of antibiotics.” However, in this report they do not look at any other factors apart from cell apart from cell size and promoter expression, but ‘many other features’ implies otherwise. This sentence should be re-phrased.

We have revised this sentence to avoid implying that we have studied many features. We also now include growth rate in this description, as these data are new with this revision.

→ *See Introduction, final paragraph*

REVIEWERS' COMMENTS:

Reviewer #1 (Remarks to the Author):

The authors have answered clearly to all points of this referee and adapted the text accordingly.

Additional experiments with the promotor-less CFP-encoding vector, and their inclusion in all downstream analyses, have made the manuscript more robust. Experiments with the *gadX* overexpression/ mutant strains show that there is not merely a correlation, but there likely is a causative link between *gadX* expression and bacterial survival under antibiotic pressure.

Reviewer #2 (Remarks to the Author):

I am satisfied that the changes and additions that the authors made to the manuscript address my previous comments. The updated analysis supports and strengthens the claims of the manuscript. Therefore, I find the current manuscript quite interesting, and a worthwhile contribution to the field. Single-cell techniques to investigate biological mechanisms present in individual cells, rather than populations, are an important tool in better understanding bacterial persistence and resistance.

I have the following minor comments:

1) In the first paragraph of the discussion the authors note: "Interestingly, the genes that had the highest predictive power were involved in ATP synthesis and/or acid resistance (*purA* *gadX* and *hdeA*) in addition to regulators involved in antibiotic resistance and oxidative stress response (*rob* and *soxS*). This highlights the coupling between responses to multiple stressors such as low pH and antibiotics." If the authors find the e.g. <https://journals.plos.org/plosgenetics/article?rev=2&id=10.1371/journal.pgen.1007284> relevant, they could put gene deletion as another stressor that implicates *gad* and *hde*. The adaptive evolution response to gene deletions like Δ *pck* also include changes to the expression of the *gad* and *hde* operons (albeit down-regulation in this reference as opposed to up-regulation that is observed in the present manuscript). These expression changes are consistent with those observed in rifamycin-resistance mutations at the RNA polymerase active site. These findings are another example of the "overlapping mechanisms cells use to cope with stress."

2) In the "rrnB P1" panel of Fig S2B, '200' should be changed to '2000'.

Reviewer #3 (Remarks to the Author):

In this revised manuscript Rossi et al have done a good job addressing the scientific concerns raised by the reviewers. The revisions have also improved the readability of the manuscript and clarity of figures, and the additional important details in the methods section will make the experiments easier for others to reproduce.

The additional experiments with overexpression of *gadX* certainly strengthen the conclusions. It is interesting that the *gadX* deletion strain shows no difference in survival to the wildtype - for future work regarding this phenomenon it would be interesting to identify conditions/stresses (e.g. acid stress) where the *gadX* deletion strain is killed faster than the wildtype, since their results suggest that such conditions should exist.

Public Repositories

Now that the manuscript is in near final form we have:

- *Uploaded the raw data sets and analysis code to a public repository. We note the exact location in the Data Availability statement.*
- *Deposited the reporter plasmids with AddGene. This is now noted in the Methods section.*

Reviewer #2 (Remarks to the Author):

1) In the first paragraph of the discussion the authors note: “Interestingly, the genes that had the highest predictive power were involved in ATP synthesis and/or acid resistance (purA gadX and hdeA) in addition to regulators involved in antibiotic resistance and oxidative stress response (rob and soxS). This highlights the coupling between responses to multiple stressors such as low pH and antibiotics.” If the authors find the e.g.

<https://journals.plos.org/plosgenetics/article?rev=2&id=10.1371/journal.pgen.1007284> relevant, they could put gene deletion as another stressor that implicates gad and hde. The adaptive evolution response to gene deletions like Δ ppk also include changes to the expression of the gad and hde operons (albeit down-regulation in this reference as opposed to up- regulation that is observed in the present manuscript). These expression changes are consistent with those observed in rifamycin-resistance mutations at the RNA polymerase active site. These findings are another example of the “overlapping mechanisms cells use to cope with stress.”

We have included this reference in the first paragraph of the Supplementary Note. Because the reference does not relate directly to antibiotic resistance we did not feel that the Discussion was the right place for it, but it fits well in the Supplementary Note.

→ See Supplementary Text, end of first paragraph

2) In the “rrnB P1” panel of Fig S2B, ‘200’ should be changed to ‘2000’.

Fixed.

→ See Fig. S2